# Improving the Screen Exploration of Smartphones Using Haptic Icons for Visually Impaired Users

**DOI:** 10.3390/s21155024

**Published:** 2021-07-24

**Authors:** Francisco Javier González-Cañete, José Luís López-Rodríguez, Pedro María Galdón, Antonio Diaz-Estrella

**Affiliations:** ETSI Telecomunicación, Departamento de Tecnología Electrónica, Universidad de Málaga, 29071 Málaga, Spain; josel.lopez@uma.es (J.L.L.-R.); mva@uma.es (P.M.G.); adiaz@uma.es (A.D.-E.)

**Keywords:** haptic icons, visually impaired, smartphones, vibration patterns

## Abstract

We report the results of a study on the learnability of the locations of haptic icons on smartphones. The aim was to study the influence of the use of complex and different vibration patterns associated with haptic icons compared to the use of simple and equal vibrations on commercial location-assistance applications. We studied the performance of users with different visual capacities (visually impaired vs. sighted) in terms of the time taken to learn the icons’ locations and the icon recognition rate. We also took into consideration the users’ satisfaction with the application developed to perform the study. The experiments concluded that the use of complex and different instead of simple and equal vibration patterns obtains better recognition rates. This improvement is even more noticeable for visually impaired users, who obtain results comparable to those achieved by sighted users.

## 1. Introduction

Our interaction with mobile phones has changed in the last few years. First, that interaction was conducted using physical keyboards integrated into the phone; since though, those keyboards have been replaced by on-screen keyboards, and our interaction with applications is currently made using touchscreens. In fact, the touchscreen is the only way to control modern smartphones. This is a great handicap for visually impaired (VI) people in terms of accessibility and usability [1], mainly because there are no physical keys or reference points, which makes it difficult to reach a specific area of the screen or activate a certain function quickly [2].

Smartphones’ user interfaces (UIs) are composed of many visual elements (icons, buttons, bars, etc.) that allow for a fast understanding of the content structure with a quick view. This is a parallel process that allows quick reading (skimming) of the whole content of the UI [3]. However, VI people lack a quick overview of the screen, and hence, they have to explore the screen randomly or use sequential techniques to scan the whole UI in order to make a mental model of the location of the items of the UI. Consequently, VI people use a much slower process to operate a smartphone [4], one requiring a high cognitive load [5,6], which has a negative impact on the user experience.

In the last few years, some researchers have been developing accessible UI for VI people (accessibility-driven blind-friendly user interfaces) [7,8,9,10,11,12,13], and some solutions have been proposed to explore the touchscreen with the aim of improving the user experience. Those proposals can be classified into three types: screen readers, logical partitions with adaptive UIs and vibrotactile feedback.

### 1.1. Screen Readers

Screen readers are the most widely used technique to manipulate a smartphone screen using a set of predefined gestures. When a finger swipes the screen to navigate sequentially over a list or matrix of elements, the system reads out (using text-to-speech) the name of each element on the list when the finger is over it. Since 2009, there have been commercial applications such as VoiceOver for iOS [14] and TalkBack for Android [15] that allow this type of interaction.

### 1.2. Logical Partitions

This technique uses logical partitions, ordered menus, guiding techniques (access overlays) and adaptive UI to improve the user experience. With the aim of accelerating navigation tasks, logical partition techniques have been proposed to organize the UI elements in a predefined or customizable scheme [16,17], to use alphabetically ordered menus [18], to form adaptive UIs that change according to the context and user preferences [1] or to form guidance techniques with access overlays [8] based on physical reference points (edge projections), the voice (neighborhood browsing and touch and speak) or data sonification. Edge projection locates all the elements on the edges of the touchscreen. The neighborhood browsing technique reads out the distance to the nearest element on the screen as the user touches it. Finally, in touch-and-speak, users have to touch the screen and say aloud system commands to obtain the direction to an objective element on the screen.

Other methods have been proposed using sounds to guide users via auditory scroll bars [19] or using stereo sound in order for the user to remember where an element is located in the menu [20] or to support them with learning gestures [21].

A constraint of the techniques based on voice or data sonification is that they are not suitable for certain environments such as noisy locations or moments that require some kind of intimacy [22].

### 1.3. Vibrotactile Feedback

Vibrotactile interaction is widely employed on touchscreen-based smartphones to generate alerts and messages and to support a secondary feedback channel that highly improves the user experience [14,23], allowing users to perform common tasks much faster (scrolling, inputting, etc.) [24,25,26]. This technique is widely used on current screen readers [27]. In addition, new technologies allow users to develop a great variety of distinguishable vibrotactile patterns that can be used to identify icons without seeing or hearing them, which is suitable for visually impaired people [28,29,30].

The research in [31] used only vibrations to move across a list and demonstrated that it helped users to memorize the list order. The results showed that the method obtained improvements compared to VoiceOver in terms of the selection time, error rate and user satisfaction. The study published in [32] combined data sonification and vibrations in order to allow visually impaired people to learn the spatial layout of the graphic information shown on a screen. The work in [33] used haptic and speech feedback to build a digital map over a touchscreen in order to make it more accessible for VI people. The study in [22] proposed a logical partition of the user interface where each partition generated a different vibration pattern.

In previous work [29], we demonstrated that the use of different vibrotactile patterns assigned to mobile applications alerts in conjunction with reinforcement in the learning process of the association improved the alert recognition rate. Besides this, we found that the improvement was more significant for VI users.

As an evolution of our previous work, the study presented in this work uses a screen reader that makes a logical partition of the touchscreen similar to the one presented in [22], but instead of using areas of the screen, we use icons with different assigned vibration patterns. Hence, the aim of this work is to assess if the alert recognition improvement obtained by using different vibrotactile patterns is also applicable to the learning of the location of icons on a tactile screen.

With the aim to assist users in learning the location of a set of haptic icons on a smartphone screen, a mobile application has been developed.

In this study, a mobile application was designed and developed with the aim of assisting users in the process of learning the locations of a set of haptic icons scattered on a smartphone screen.

Considering the literature reviewed, we stated our hypotheses as:The use of complex vibration patterns (vibration patterns with different intensities, durations and numbers of pulses) associated with smartphone application haptic icons instead of simple and equal vibrations (the same vibration pattern that consists of a simple vibration pulse) improves the user experience in terms of the recognition ratio and the memorization of the haptic icons’ locations.This improvement also applies to visually impaired users.

## 2. Materials and Methods

### 2.1. Participants

Forty-six participants aged 18–60 years old took part in the experiments. Eighteen of them had a visual disability ranging from 80% to 100%, and consequently, they were unable to use graphical user interfaces. However, they used accessibility applications such as the Android Accessibility Suite to be able to interact with their smartphones. They were volunteers that worked in ONCE (Organización Nacional de Ciegos Españoles—Spanish National Blind Association). The only requirement was that they had to be smartphone users in order to be sure they knew how to use them as well as their accessibility applications. On the other hand, 28 of the participants were fully sighted, non-visually impaired users. They were student and teacher volunteers from the Electronic Technology faculty of the University of Málaga.

### 2.2. Experimental Design

The users’ visual capacities (VI vs. sighted) and the vibration types of the haptic icons (simple and equal vs. complex and different vibration patterns) were considered as independent variables in a 2 × 2 design. On the other hand, we considered the time needed to solve the test, the haptic icon recognition rate and the evaluation of the usability questionnaire as dependent variables.

All participants participated in a previous learning stage in which they navigated the screen and received acoustic information of the touched haptic icon (in the form of a voice telling them the name of the related application) while the vibration related to the pressed haptic icon was played.

Half of the participants (23) used the developed application with simple vibrations assigned to each haptic icon (14 sighted and 9 VI) and the other half used the application with complex vibrations assigned to each haptic icon (14 sighted and 9 VI). Table 1 summarizes the subjects’ distribution.

Informed written consent was given by all participants in the study. The data obtained were analyzed anonymously.

### 2.3. Stimuli and Devices

An Android application named EXT (Enhanced eXplore by Touch) was developed as a tool to implement the experiments. The source code can be freely accessed online (https://github.com/Equinoxe-fgc/EXT (accessed on 23 July 2021)). The EXT haptic icons were chosen from those predefined in the gallery of the Haptic Effect Preview tool [34]. No conceptual meanings could be associated with the applications they represented (Figure 1).

To perform the tests, a Samsung Galaxy S3 smartphone was employed because this is one of the recommended models to use with the Immersion SDK for Mobile Haptics [35]. As the vibrations were in the audio range of frequencies, they were recorded using a microphone attached to the smartphone and post-processed in order to obtain a good quality graphical representation of the vibrations. After analyzing the recorded waves, the vibration frequency was determined to be 200 Hz. Consequently, a bandpass filter from 100 to 300 Hz was applied to the signals to remove artifacts caused by screen touches or environmental noise. Afterward, the amplitudes were normalized in order to be comparable. The signal waveforms of the EXT haptic icon are shown in Figure 2, where the y-axis represents the relative intensity, and the x-axis is measured in seconds. The audio vibration patterns are available as Appendix A.

According to the type of vibration, the utilized haptic icon can be categorized into seven classes:Single click: They are very short vibrations (0.1 s). The vibration is so short that the vibration intensity does not have enough time to reach the maximum specified value. They are used by the Internet, Facebook and Twitter haptic icons.Double-click: They are two consecutive short vibrations. In the case of the SMS haptic icon, both vibrations do not overlap as the first vibration ends before the second one starts. On the contrary, the Clock and Play Store haptic icon vibrations do overlap, with the difference between them equal to the time between clicks.Triple-click: It is composed of three consecutive, overlapping clicks (Hangouts).Buzz: They are symmetrical vibrations that hit the peak of the effect’s vibration amplitude somewhere in the middle, sustaining it for a period of time. Calculator and LinkedIn use this type of vibration with different amplitudes and durations.Ramp: They vary the vibration amplitude, either increasing it over time (ramp-up) or starting with a high amplitude and decreasing it gradually until it stops (ramp-down). Google+ uses a ramp-up vibration with a rise time of 0.5 s and a release time of 0.15 s. Meanwhile, WhatsApp uses a ramp-down with a very short rise time of 0.08 s and a release time of 0.5 s.Pulse: They are smooth ramps up and down combined. The Email haptic icon uses three consecutive pulses of 0.5 s for each one.Buzz-bump: They are haptic icons with a combination of a buzz and a bump. A bump is a softer click. Calendar and Line use the same buzz but a different bump duration, with the Calendar bump having double the duration to the one used by Line. Calendar, Phone and Camera share the bump characteristics but differ on the bump intensity.

### 2.4. Experimental Procedure

As a previous step to the experiment, the subjects had to memorize the locations of sixteen icons related to applications and their associated vibrotactile stimuli. The aim of the experiment was to establish whether the use of different and complex vibrations had a better success ratio in terms of the users’ ability to memorize the locations of haptic icons on the smartphone screen. Additionally, the experiment sought to conclude if the improvement applied to visually impaired people. If it could be demonstrated, this improvement could be employed to increase smartphone accessibility and navigation for people with visual disabilities. Accessibility applications such as TalkBack for Android-based smartphones use simple and equal vibrations for all the icons shown on-screen. Hence, these could be improved using different and complex vibrations if it can be demonstrated that this type of vibration for haptic icons obtains a better recognition rate.

The application is divided into two sections:Training section: In this section, the screen shows sixteen different haptic icons in a 4 × 4 grid. Every icon corresponds to an application (Facebook, Line, Camera, etc.). When an icon is pressed, the smartphone plays a recording with the application name, and the vibration associated with the haptic icon is also played (Figure 3). The user can have as many trials and time as they would like to learn the locations and associated vibration patterns of all haptic icons.Test section: This section shows the same sixteen haptic icons as the training section, with the same distribution, although they are hidden by “?” symbols (Figure 4). The subject can explore the screen, touching it to find the icons. When a haptic icon is touched, a ‘beep’ sound is played. The menu key of the smartphone is programmed to play the name of the application the subject must find on the screen, which they are to do in the quickest amount of time possible. When the subject has selected the haptic icon to use as a response, the EXT application informs them whether the answer is correct. If it is correct, the application’s visual icon is shown and remains visible for the remainder of the test. In addition, if the icon is touched again, a voice will be played with the name of the application. However, if the answer is wrong, the application’s visual icon will be hidden again by the icon with the “?” symbol and the user will have to continue searching for the requested icon without any limit on the number of attempts, although this particular icon’s recognition will be considered as failed. Every time a haptic icon has been correctly found, the name of the next application to be found will be played until the sixteen icons are located.

The application allows two modes of operation: using simple or complex vibrations patterns. This feature will allow us to determine if the use of complex haptic icon vibrations improves the users’ capability to remember the locations of icons on the smartphone screen compared with the use of simple vibration patterns used by accessibility applications such as TalkBack.

Simple vibration patterns: Every haptic icon uses the same simple vibration pattern to provide feedback to the subject when exploring the smartphone screen using the sense of touch, as with the TalkBack accessibility application. The Facebook vibration pattern (Figure 2) was selected for this as it is a short and intense vibration.Complex vibration patterns: Every haptic icon uses different and complex vibration patterns, as shown in Figure 1 and Figure 2.

After the test was completed, all the subjects had to fill out a six-question usability questionnaire. The first three questions were adapted from the System Usability Scale (SUS) and Computer System Usability Questionnaire (CSUQ) [36] usability forms. The last three questions were related to the perception, differentiation and recognition [37] of the haptic icons of the developed application. A seven-level Likert scale was given, ranging from totally disagree (value 1) to totally agree (value 7). Table 2 shows the questions used for evaluation.

## 3. Results

This section presents the results of the experiments, focusing on the statistically significant results (*p* < 0.05).

### 3.1. Test Time

The main descriptive statistics of the test time differentiating VI and sighted subjects are shown in Table 3. The mean execution time of the test section was 7 min 22 s (5 min 52 s for sighted subjects and 9 min 42 s for VI subjects). The minimum test time for both types of subjects was 3 min; moreover, the maximum test time was 12 min for sighted subjects, but this was double, 24 min, for VI subjects. The mean test time needed for those subjects that used simple vibrations was 8 min 30 s, while the mean time taken when using complex vibrations was 6 min 14 s.

A two-way ANOVA (ANalysis Of VAriance) was conducted to examine the effect of the visual capacity and the type of vibration on the time taken to perform the test (Table 4).

There was no statistically significant interaction between the effects of both parameters on the time taken (F (2.46) = 0.465, significance = 0.499 > 0.05). However, simple main effects analysis showed that the vibration type significantly affects the test time (F (1.46) = 4.672 and significance = 0.036 < 0.05). The same behavior is observed regarding the visual capacity of the subject, which also affects the recognition time (F (1.46) = 11.583, significance = 0.01 < 0.05). In fact, the visual capacity has the main influence on the test time as VI subjects need more time than sighted subjects to complete the test.

### 3.2. Recognition Rate

The two-way ANOVA test shows a statistically significant interaction between the effects of the visual capacity of the subjects and the type of vibration (simple or complex) on the recognition rate (Table 5; F (2.54) = 8.397, significance = 0.006 < 0.05).

Simple main effects analysis shows that significant differences exist between subjects with different visual capacities (F (2.46) = 15.15, significance = 0.0 < 0.05) and the type of vibrations (F (2.46) = 30.96, significance = 0.0 < 0.05).

As shown in Figure 5, the recognition rate is considerably improved when using complex and different vibration patterns for VI subjects, increasing from 0.47 to 0.83. Moreover, the recognition rate is slightly increased (0.12) for sighted subjects when using complex vibrations. Consequently, it can be deduced that the use of different and complex vibration patterns associated with the icons of applications significantly improves (0.36) the recognition rate for VI users, while it slightly increases the recognition rate of sighted subjects. This conclusion agrees with the results obtained by the one-way ANOVA test of the influence of the use of simple or complex vibrations on VI (F (1.18) = 29.641, significance = 0.0 < 0.05) and sighted (F (1.28) = 4.521, significance = 0.043 < 0.05) subjects.

The recognition rate per haptic icon as a function of the type of vibration pattern (with unique and simple vibrations compared to complex and different vibrations) and without, considering the type of subject (VI or sighted), is shown in Figure 6.

The recognition rate increases for all haptic icons when using complex vibration patterns compared to the results obtained using simple vibrations. This is the expected behavior as complex vibrations have more information than simple vibrations. However, in spite of this predictable result, assistive applications such as TalkBack for Android do not use this advantage. The most noticeable improvements are for the Internet (reaching 1.0), Facebook and Twitter, which have single-click vibrations with different intensities and durations, as well as the icons with pulses and buzz-bump vibrations. Considering the single-click vibrations, it is expected that these will be more distinguishable from the simple vibrations (using the Facebook vibration pattern) as it is a way to differentiate them. On the other hand, the buzz-bump vibration patterns (Calendar, Line, Phone and Camera) are very different from the rest of the vibration patterns, hence, they are expected to be more discernable. Finally, the Email haptic icon also improves as the associated pulse vibration pattern is the most different from the rest of the vibration patterns in terms of duration and structure.

Figure 7 compares the recognition rate of sighted subjects using simple and complex vibration patterns. The figure shows that for most of the haptic icons, the recognition rates are slightly increased except for the Internet, Facebook and Twitter, where the increase is more noticeable, as shown in Figure 6. The LinkedIn (buzz) and Email (pulses) haptic icons also perform much better. However, the SMS (double-click) and Google+ (ramp-up) icons slightly reduce in their recognition rates.

The same study (results presented in Figure 7) is repeated but considering the recognition rate per haptic icon of the VI subjects as a function of the use of simple or complex vibration patterns, with the results shown in Figure 8. It can be observed that the recognition rate is considerably increased when using complex vibration patterns, especially for the SMS, Calendar and Camera haptic icons. It is worth mentioning that the Calendar haptic icon obtained only a 0.2 recognition rate when using simple vibrations and obtained a 1.0 recognition rate (it is always recognized) when using complex vibrations. In previous work [29], some VI subjects pointed out that they had more problems recognizing the haptic icons when they were associated with applications they did not use regularly. Consequently, it is expected that applications such as SMS (that has marginal use nowadays) or the Camera will obtain low recognition rates for VI people.

Figure 9 compares the haptic icon recognition rates of VI and sighted subjects when using complex and different vibration patterns. It can be observed that the recognition rates obtained by VI subjects are very close to or even better than those obtained by the sighted subjects for most of the haptic icons. For VI subjects, the Internet and Calendar haptic icons are fully recognized; the same behavior is observed for sighted subjects for the Internet (sharp click), Facebook (strong click), Play Store (double bump click) and Calculator (long buzz) haptic icons. Sighted people obtain better results than VI people at recognizing the Facebook (strong click), Calculator (long buzz) and LinkedIn (short buzz) haptic icons. On the other hand, VI subjects better recognize the SMS (double sharp click), Calendar (transition bump), Line (transition bounce 100) and Camera (transition bounce 33) haptic icons.

In our experiment, the haptic icons’ locations were fixed and, hence, it may be possible that the memorization of their position was related to their relative location. That is, haptic icons situated in certain locations could have been more easily remembered because of their position instead of their vibration patterns. Figure 10 represents, using a heatmap graph, the same information shown in Figure 9 but taking into consideration the haptic icons’ positions. The center of the figure shows the haptic icons’ locations, while the left and right graphs show the recognition rates for VI and sighted people, respectively, using complex vibration patterns. The colors represent, according to the legend shown in the right part of the figure, the recognition rate, using colors close to green for values close to 1.0 and red for values those close to 0.0. The displayed values for VI people do not show any pattern that could infer that there is a relationship between the recognition rate and the haptic icon location. Nevertheless, sighted people seem to better recognize or remember the first row of haptic icons. The Internet haptic icon is perfectly recognized by both types of users because it is the first one and it is thus easier to remember.

Finally, Figure 11 represents the recognition rates of VI and sighted people considering the types of vibration patterns classified in Section 2.3., that is, click, double-click, triple-click, buzz, ramp, pulse and buzz-bump vibration. There was no significant difference between the VI and sighted users’ performances. Sighted users usually obtained a better recognition rate, except for pulses and buzz-bumps, for which VI users outperformed sighted users, but these variations were insignificant. The noticeable difference was for buzz vibrations (Calculator and LinkedIn), where sighted users outperformed VI people by more than 0.20. Unfortunately, with the collected data, we cannot infer which vibration was selected when an icon was incorrectly recognized, only whether it was recognized.

### 3.3. Usability Form

We implemented nonparametric Mann-Whitney U tests to compare the replies on the use of simple and equal vibration patterns or complex and different ones for the VI and sighted subjects on the usability form.

Table 6 shows the usability answers from the EXT application. There is a clear difference between sighted and VI subjects, as sighted people stated that vibrations were not clearly perceived, while VI people answered that they were very clearly perceived (Question 1). This difference is the same for simple and complex vibrations. However, all the subjects perceived that the vibrations were not clearly distinguishable (Question 2) or that it was not easy or was difficult to recognize the meaning of each haptic icon (Question 3), even though the recognition rates were high, especially when using complex vibration patterns. According to the answers, VI people would like to assign vibrations to assist their tactile exploration of the mobile phone screen (Question 4), although sighted subjects would not take up this option. All the sighted subjects affirm that they would need more practice to remember the icons’ locations (Question 5). Finally, all the subjects agree that the EXT application is very easy to use (Question 6).

## 4. Discussion

In this paper, we have presented a group of experiments with the aim to evaluate the learning of the locations of a set of haptic icons on a smartphone screen. Considering the time taken to learn the icons’ locations, it was noticeable that they could be learned in less than ten minutes, even though this time was different depending on the visual capacity of the subjects.

### 4.1. Effect of the Vibration Type

Our first hypothesis was that complex vibration patterns associated with smartphone application haptic icons could improve the user experience in terms of icon location on a smartphone screen. The experiments showed that the time needed to memorize and remember the icons’ locations was reduced compared to the use of simple and equal vibrations. In addition, the experiments also demonstrated that the icon location recognition rate was also improved, as the subjects considered it easier to associate different locations with different vibrations. This improvement was consistent with the one obtained in our previous work [29], which demonstrated that the use of different vibration patterns also improved the recognition of alerts associated with mobile applications.

However, assistance applications such as TalkBack for Android do not use different vibration patterns but rather use simple and equal ones. Hence, it would be a great improvement in terms of usability to adapt this type of application by using different vibration patterns and even allowing the users to associate the vibrations with certain application icons.

Considering the types of vibration patterns, the single-click applications (Internet, Facebook and Twitter) are much better distinguished than when all of them use simple vibrations (the Facebook waveform vibration). The same behavior is noticed for buzz-bump and pulses vibration patterns.

### 4.2. Effect of the Subjects’ Visual Condition

Our second hypothesis was that the possible improvements obtained by using complex and different vibration patterns also applied to visually impaired users. The conducted experiments showed that the VI subjects had a drastically reduced time taken to learn the icons’ locations compared to the time taken by sighted subjects. This behavior suggests that the learning process is highly supported by the different vibrotactile feedback obtained. In addition, the icon location recognition rate was also highly improved when using different and complex vibration patterns, allowing VI subjects to obtain similar results to sighted subjects who can also use their vision to learn the icon locations. This improvement was even more noticeable for some types of applications (those that are not usually employed by VI people). Considering the recognition rate as a function of the type of vibration patterns, the results are consistent with the previously stated results, and hence, the selected vibration patterns do not significantly affect the results. However, a deeper study of the distinguishability of the selected vibration patterns could be performed as a future work.

Consequently, the use of complex and different vibration patterns associated with application icons is highly recommended for VI users, as it allows them to achieve a user experience very similar to that of sighted users.

## 5. Conclusions

In this paper, we proposed the use of complex and different vibration patterns and associated them with different haptic icons on a smartphone, which is in contrast to the use of simple and equal vibrations as performed by commercial assistance applications. The experiments conducted considered the visual capability of the subjects in order to study whether the user experience of visually impaired users could also be improved when using smartphone devices.

An Android smartphone application was developed to perform the experiments. The application consists of two stages: a learning stage where users memorize the locations of haptic icons and the associated vibrations, and a test stage where users recognize the locations of haptic icons using vibrotactile feedback. The application tracks the time taken by subjects to learn the locations of the haptic icons presented on the screen and the recognition rate of each haptic icon.

The obtained results show that the use of different and complex vibration patterns associated with haptic icons reduces the time needed to learn the icons’ locations. This reduction is more noticeable for VI people. The recognition rate is also improved compared to the results obtained by using simple and equal vibration patterns. The improvement obtained by VI users is so great that these users demonstrated almost the same recognition rate as that of sighted users. This leads us to conclude that the use of complex vibration patterns improves the user experience, especially for VI people. It can also add new features for sighted people as they can easily navigate “non-visually” on their phone while it is in their pocket, in a backpack or while looking at another screen.

For future work, we propose testing the EXT application using different smartphone models to determine whether the recognition rate changes, as not all the smartphone models have the same vibration accuracy and capacity. Some users proposed that we add a new feature to the application in terms of the capability to configure the vibration patterns to the applications. Consequently, a new experiment could be performed in order to measure whether the recognition rate improves. Another improvement to the EXT application would be to store information about the user selection when a haptic icon is incorrectly located. This could add information about which vibration patterns are confused.

## Figures and Tables

**Figure 1 sensors-21-05024-f001:**
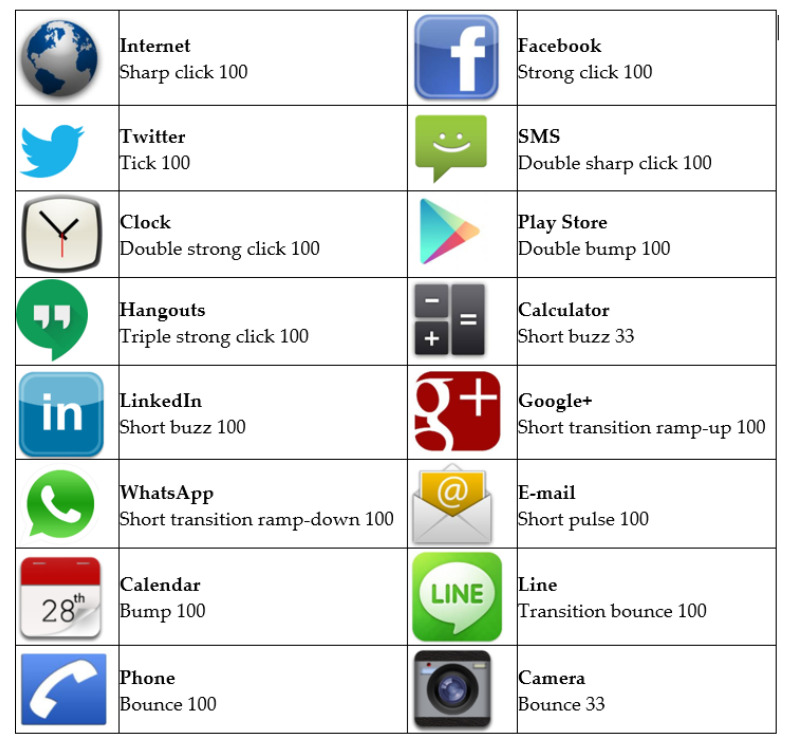
Haptic icons and their related vibrations. The vibrations have been chosen from the Haptic Effect Preview application. The numbers represent the vibration intensities as percentages with 100 being the maximum vibration intensity.

**Figure 2 sensors-21-05024-f002:**
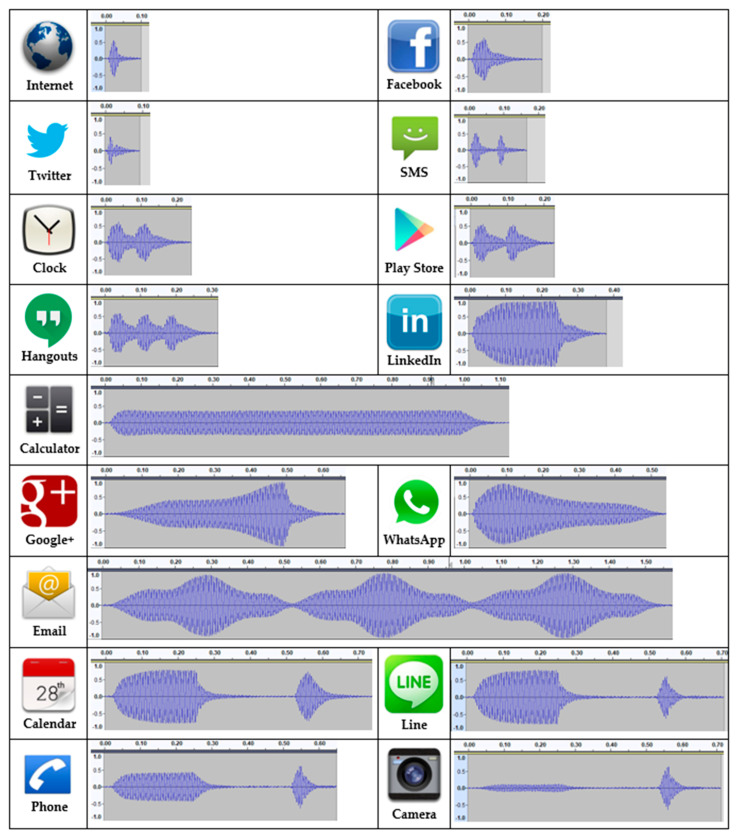
Haptic icons and their related vibration waveforms.

**Figure 3 sensors-21-05024-f003:**
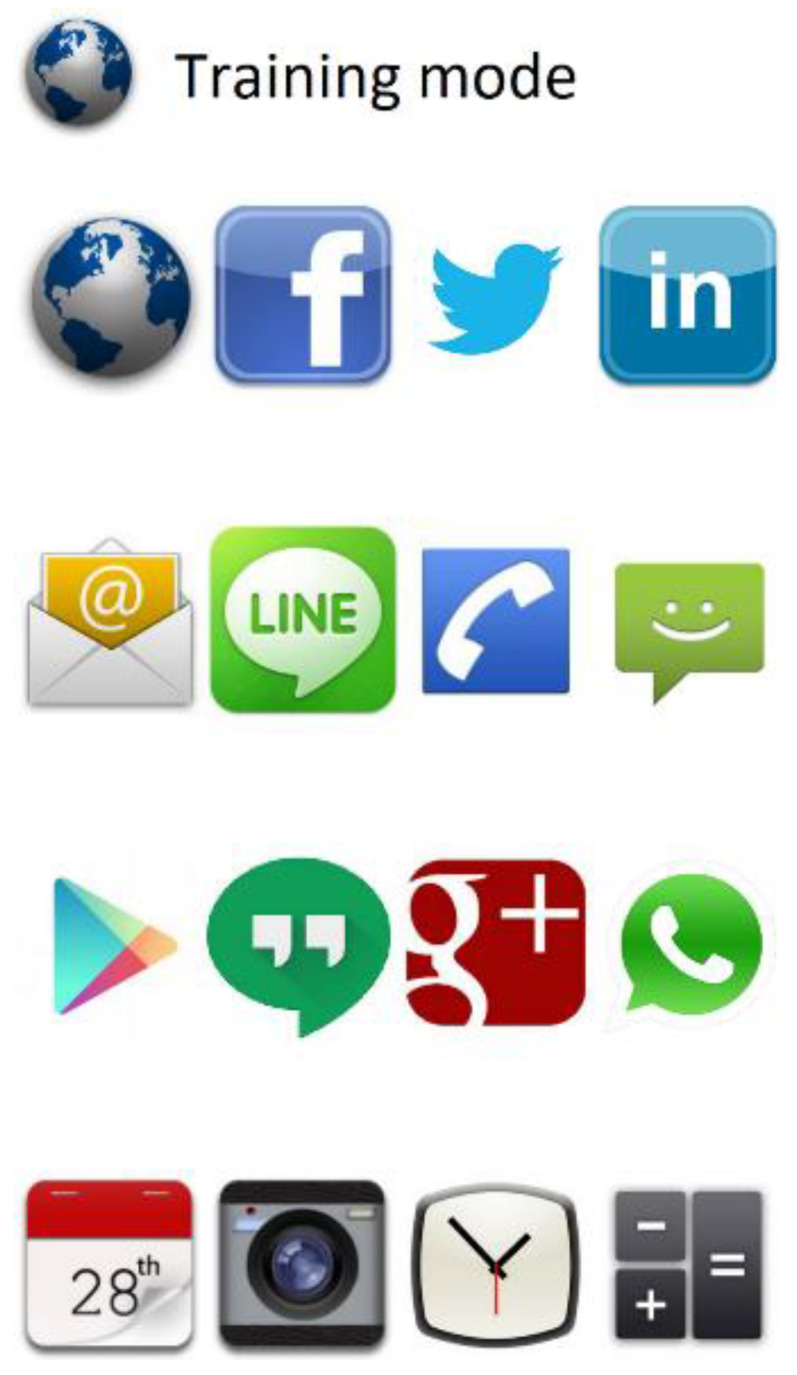
Training section screen of the EXT application. A set of sixteen haptic icons is shown in a 4 × 4 grid.

**Figure 4 sensors-21-05024-f004:**
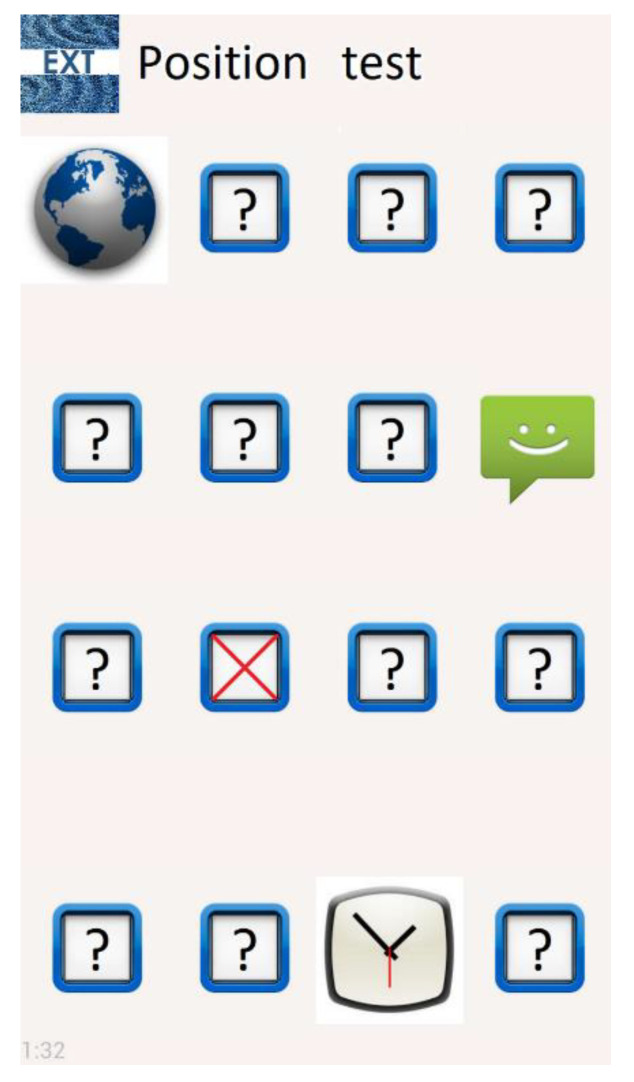
Test section screen. The visible icons have been correctly recognized and the “?” icons hide the haptic icons to be recognized. The numbers on the bottom-left corner measure the time taken to make the test.

**Figure 5 sensors-21-05024-f005:**
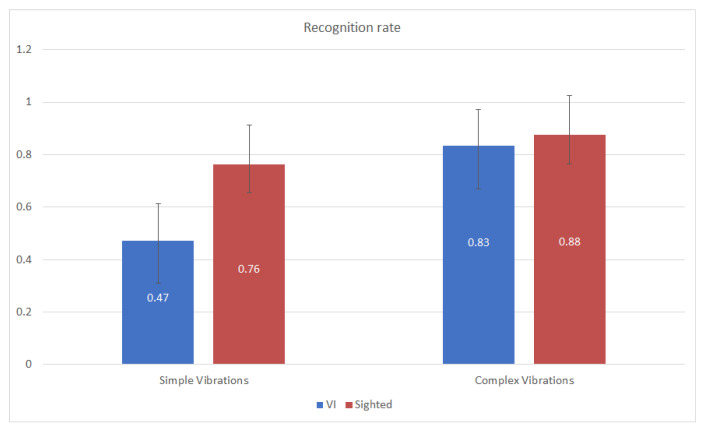
Application recognition rate. For all the subjects, this figure shows the recognition rate as a function of the use of simple and equal vibrations patterns or complex and different vibration patterns while differentiating between VI and sighted subjects.

**Figure 6 sensors-21-05024-f006:**
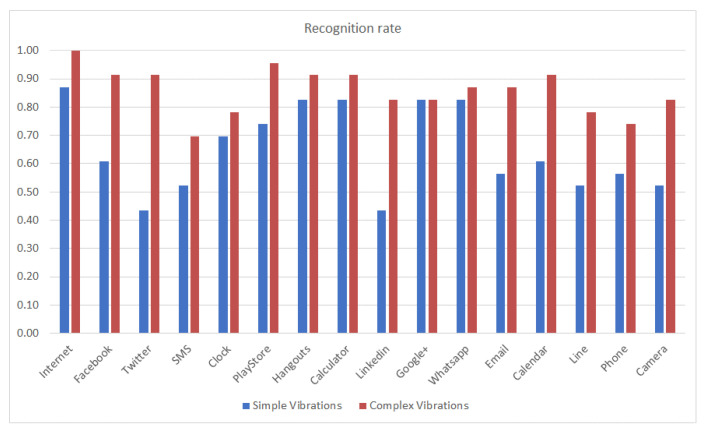
Recognition rate per haptic icon for simple vs. complex vibrations for the whole population.

**Figure 7 sensors-21-05024-f007:**
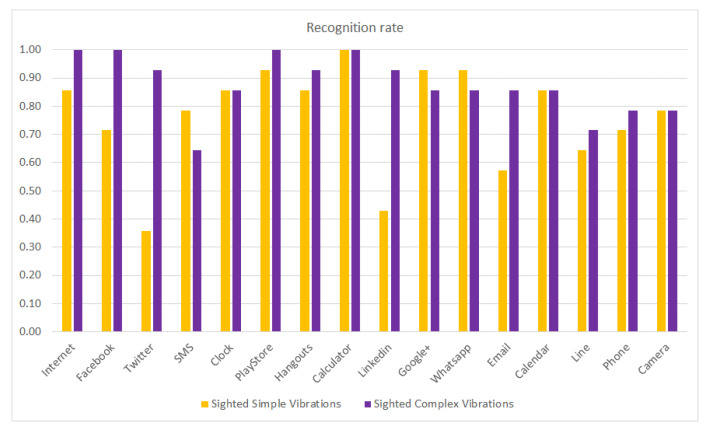
Recognition rate per haptic icon for sighted people using simple and complex vibration patterns.

**Figure 8 sensors-21-05024-f008:**
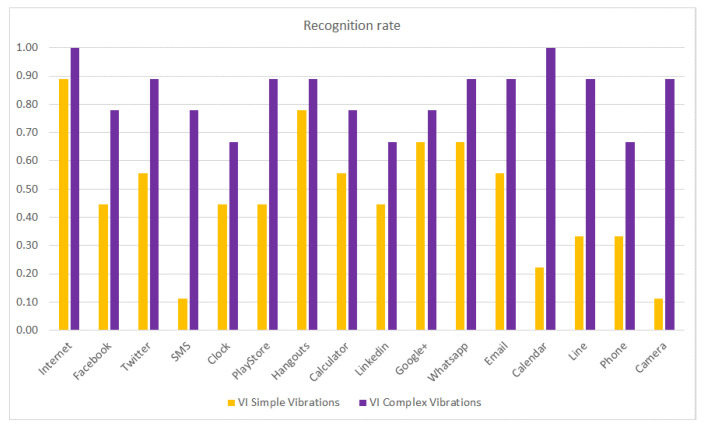
Recognition rate per haptic icon for VI people using simple and complex vibration patterns.

**Figure 9 sensors-21-05024-f009:**
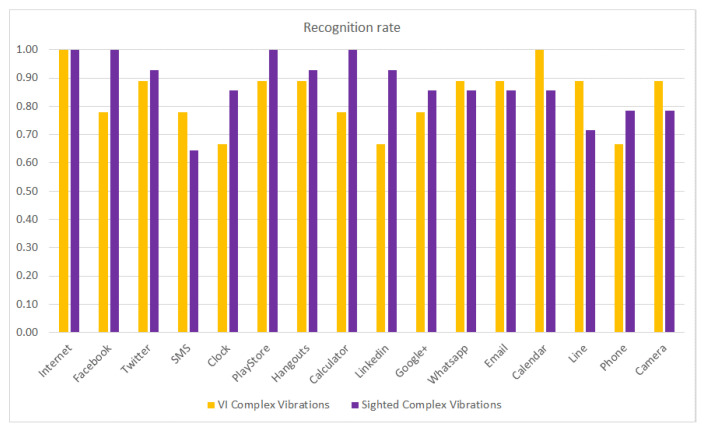
Recognition rate per haptic icon for VI and sighted people using complex vibration patterns.

**Figure 10 sensors-21-05024-f010:**
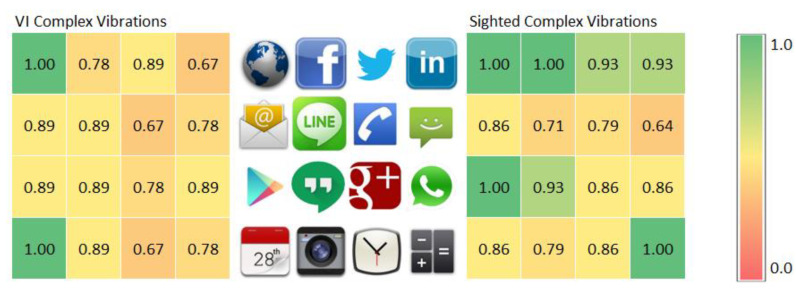
Heatmap recognition rate per haptic icon for VI (**left**) and sighted (**right**) people using complex vibration patterns. The haptic icons’ locations are represented in the (**center**) of the figure.

**Figure 11 sensors-21-05024-f011:**
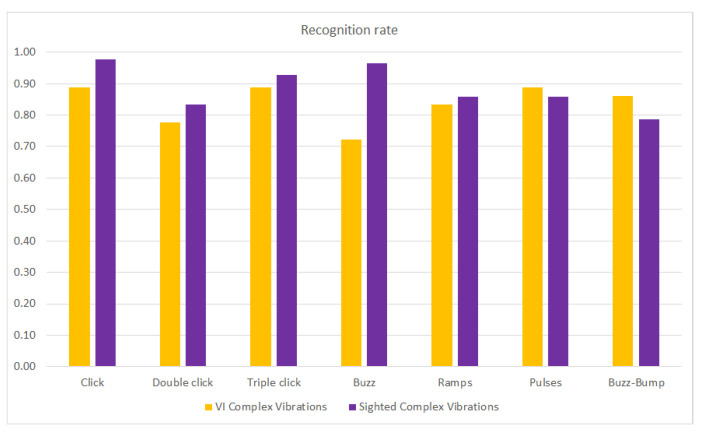
Recognition rate per type of vibration pattern for VI and sighted people.

**Table 1 sensors-21-05024-t001:** Distribution of subjects according to their visual capacity and the vibration type used in the EXT application.

		*n*
Visual Capacity	Sighted	28
	Visually Impaired	18
Vibration Type	Simple vibrations	23
	Complex vibrations	23

**Table 2 sensors-21-05024-t002:** Usability form.

Question Number	Question
1	Were the vibrations clearly perceived?
2	Were the vibrations distinguishable?
3	Was it possible to recognize the meaning of each haptic icon?
4	Would you like to assign vibrations to assist the tactile exploration of the mobile phone screen?
5	Will you need more practice to remember all the vibration patterns?
6	Is the EXT application easy to use?

**Table 3 sensors-21-05024-t003:** Descriptive statistics for the test time (in minutes:seconds), grouping VI and sighted subjects as a function of the vibration patterns used.

Visual Capacity	Vibration Type	Mean	Standard Deviation	*n*
Sighted	Simple vibrations	6:42	2:59	14
	Complex vibrations	5:03	1:08	14
	Total	5:52	2:22	28
Visually Impaired	Simple vibrations	11:18	6:20	9
	Complex vibrations	8:06	3:58	9
	Total	9:42	5:23	18
Total	Simple vibrations	8:30	5:01	23
	Complex vibrations	6:14	2:58	23
	Total	7:22	4:14	46

**Table 4 sensors-21-05024-t004:** Two-way ANOVA study of the time test as a function of the visual capacity of the subjects and the vibration type used (being df the degrees of freedom).

Source	Type-III Sum of Squares	df	Mean Square	F	Significance
Visual Capacity	578,001.104	1	578,001.104	11.583	0.001
Vibration Type	233,143.133	1	233,143.133	4.672	0.036
Visual Capacity × Vibration Type	23,200.003	1	23,200.003	0.465	0.499
Error	2,095,863.937	42	49,901.522		
Total	11,928,843.000	46			
Corrected Total	2,909,361.239	45			

R-squared = 0.280 (adjusted R-squared = 0.228).

**Table 5 sensors-21-05024-t005:** Two-way ANOVA test of the recognition rate as a function of the visual capacity of the subjects and the vibration type used.

Source	Type-III Sum of Squares	df	Mean Square	F	Significance
Visual Capacity	0.291	1	0.291	15.158	0.000
Vibration Type	0.594	1	0.594	30.962	0.000
Visual Capacity × Vibration Type	0.161	1	0.161	8.397	0.006
Error	0.806	42	0.019		
Total	28.000	46			
Corrected Total	1.749	45			

R-squared = 0.539 (adjusted R-squared = 0.506).

**Table 6 sensors-21-05024-t006:** Median results of the EXT application usability form.

	Simple Vibrations		Complex Vibrations		
Question	Sighted (*n* = 14)	VI (*n* = 9)	Sighted (*n* = 14)	VI (*n* = 9)	Total (*n* = 46)
1	2.5	7	2	7	6
2	1	2	1	1	1
3	1	4	1	2	1
4	1	4	2	4	2
5	6.5	5	4	2	5
6	7	7	7	7	7

## Data Availability

The data presented in this study are available on request from the corresponding author. The data are not publicly available due to privacy.

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
