# Peer review of "Improving the Screen Exploration of Smartphones Using Haptic Icons for Visually Impaired Users"

_sensors, 2021, doi:10.3390/s21155024_

Round 1
Reviewer 1 Report
The paper introduces a comparative study about the design of haptic icons. The major factor examined in the study was a variable set of complex haptic icons vs. simple vibration, and the result indicates varying the haptic signals is beneficial to a discrimination of the menu items on a touchscreen device. However, I feel the manuscript is missing a massive amount of details. 1. First of all. the haptic icons are presented with few words (e.g., short pulse 100, bounce 33, etc.) that probably came from the Haptic Effect Preview tool from a company named Immersions' proprietary asset. This prevents readers comprehend the design of the haptic vibration fully. Also, even though the tool is generating consistent vibration signals, different devices render the signals differently. Moreover, the mobile device itself was not clarified in the paper. Therefore the most important information -- which vibration was delivered to the participants -- is completely missing in this paper. A better way is presenting the signal in terms of acceleration, using some vibration measuring device (for example, accelerometer, or laser optical vibration measurement). In this way, the rendered signal could be reproduced on other devices, and even after the proprietary app is stop deployed by the company. 2. The paper concluded that the complex haptic icon is better than a simple vibration, which is way too obvious because the haptic icon in the "complex" setting simply transferred more information to a user. It's saying like that, using 8-bit per message can deliver more information than 1-bit per message -- which is a way too obvious conclusion to be discussed. A more informative way of approaching this should be, which aspect of haptic vibration contributes to the efficacy of the information delivery. In this aspect, the duration, amplitude, type, etc. of a signal should be decomposed and analyzed with the result data. However, the presented results (from Figure 5 to 10) are delivering basically no information, because a deeper analysis is missing regarding the characteristics of each signal. It's connected to the problem I indicated in the previous section 1 (missing the description of the haptic vibrations) as well. Because the two problems I mentioned are too critical, I argue this paper to be rejected. A more detailed discussion about this paper is not worthy because I expect all the details are subject to be changed in the potential revision of this paper.
Author Response
I attach the response in a PDF file

Reviewer 2 Report
In this work, the authors investigate the use of complex vibrations to support sighted and visually-impaired people in learning the positions of icons on the touch-screen. The study compares simple and complex haptic icons with the aim to improve the simple feedback provided by the commercial applications and screen readers.
The study is well-conducted and the users were grouped into two different sub-groups to compare simple and complex vibrations.
The work is simple and well written.
Just some few comments:
In the introduction, the statement “use sequential techniques” in the sentence “...However, for VI people, the only option is to use sequential techniques sentence...” is incomplete or inexact: the VI users can move sequentially or explore the screen randomly. The main concept is however in the right way: the main issue is that the user cannot have an overview of the screen. This could be better expressed in the paper.
Some additional considerations on the complex vibrations could be provided in the paper.
The study could also consider to test both simple and complex haptic icons with the same group of users rather than considering them as separate tests.
In conclusion, the work can provide a further step in the field.
Author Response
I attach the response in a PDF file

Reviewer 3 Report
This pas is discussing the issue of screen exploration for visually impaired users using haptic icons and a test application.
The paper is wells structured and in very good technical depth.
The following things need to be clarified:
- How the users were selected in terms of the tests and the degree of impairment (this must be more elaborated)
- Figures 5,6,7,8.9,10 are somehow confusing (they present positive result but not consistent) the advice is to explain them in detail and explain the differences .Consider using different types of plots to better describe the output. A summary in the discussion is also useful I order to better present the output of this work
- What and how this work is different than previous ones? How much this specific implementation is biasing the output results? Compare with other works?
Author Response
I attach the response in a PDF file

Round 2
Reviewer 1 Report
I appreciate the authors added Figure 2, which makes understanding the vibration characteristics of the system clear.
I also appreciate the elaborated analysis of the results in section 3.2, which now contains more qualitative information compared to v1 manuscript.
Overall the paper is looking good, however, I'd like to suggest the authors illustrate a better visualization of the figures in section 3.2. In my personal opinion, a bar graph is not very effective in this case, because the items were spatially arranged with graphic icons, and the bar graphs are listing them linearly only with their text labels. I think a heatmap should be a better visualization choice for these kinds of results. Please consider this in the revision.
Author Response
See the attached document
